# ARE POWERFUL GRAPH NEURAL NETS NECESSARY? A DISSECTION ON GRAPH CLASSIFICATION

## ABSTRACT

Graph Neural Nets (GNNs) have received increasing attentions, partially due to their superior performance in many node and graph classification tasks. However, there is a lack of understanding on what they are learning and how sophisticated the learned graph functions are. In this work, we propose a dissection of GNNs on graph classification into two parts: 1) the graph filtering, where graph-based neighbor aggregations are performed, and 2) the set function, where a set of hidden node features are composed for prediction. To study the importance of both parts, we propose to linearize them separately. We first linearize the graph filtering function, resulting Graph Feature Network (GFN), which is a simple lightweight neural net defined on a *set* of graph augmented features. Further linearization of GFN's set function results in Graph Linear Network (GLN), which is a linear function. Empirically we perform evaluations on common graph classification benchmarks. To our surprise, we find that, despite the simplification, GFN could match or exceed the best accuracies produced by recently proposed GNNs (with a fraction of computation cost), while GLN underperforms significantly. Our results demonstrate the importance of non-linear set function, and suggest that linear graph filtering with non-linear set function is an efficient and powerful scheme for modeling existing graph classification benchmarks.

## 1 INTRODUCTION

Recent years have seen increasing attention to Graph Neural Nets (GNNs) (Scarselli et al., 2009; Li et al., 2015; Defferrard et al., 2016; Kipf and Welling, 2016), which have achieved superior performance in many graph tasks, such as node classification (Kipf and Welling, 2016; Wu et al., 2019) and graph classification (Simonovsky and Komodakis, 2017; Xinyi and Chen, 2019). Different from traditional neural networks that are defined on regular structures such as sequences or images, graphs provide a more general abstraction for structured data, which subsume regular structures as special cases. The power of GNNs is that they can directly define learnable compositional function on (arbitrary) graphs, thus extending classic networks (e.g. CNNs, RNNs) to more irregular and general domains.

Despite their success, it is unclear what GNNs have learned, and how sophisticated the learned graph functions are. It is shown in (Zeiler and Fergus, 2014) that traditional CNNs used in image recognition have learned complex hierarchical and compositional features, and that deep non-linear computation can be beneficial He et al. (2016). Is this also the case when applying GNNs to common graph problems? Recently, Wu et al. (2019) showed that, for common node classification benchmarks, non-linearity can be removed in GNNs without suffering much loss of performance. The resulting linear GNNs collapse into a logistic regression on graph propagated features. This raises doubts on the necessity of complex GNNs, which require much more expensive computation, for node classification benchmarks. Here we take a step further dissecting GNNs, and examine the necessity of complex GNN parts on more challenging graph classification benchmarks (Yanardag and Vishwanathan, 2015; Zhang et al., 2018a; Xinyi and Chen, 2019).

To better understand GNNs on graph classification, we dissect it into two parts/stages: 1) the graph filtering part, where graph-based neighbor aggregations are performed, and 2) the set function part, where a set of hidden node features are composed for prediction. We aim to test the importance of both parts separately, and seek answers to the following questions. *Do we need a sophisticated graph*

*filtering function for a particular task or dataset? And if we have a powerful set function, is it enough to use a simple graph filtering function?*

To answer these questions, we propose to linearize both parts separately. We first linearize graph filtering part, resulting Graph Feature Network (GFN): a simple lightweight neural net defined on a set of graph augmented features. Unlike GNNs, which learn a multi-step neighbor aggregation function on graphs (Dai et al., 2016; Gilmer et al., 2017), the GFN only utilizes graphs in constructing its input features. It first augments nodes with graph structural and propagated features, and then learns a neural net directly on the *set* of nodes (i.e. a bag of graph pre-processed feature vectors), which make it more efficient. We then further linearize set function in GFN, and arrive at Graph Linear Network (GLN), which is a linear function of augmented graph features.

Empirically, we perform evaluations on common graph classification benchmarks (Yanardag and Vishwanathan, 2015; Zhang et al., 2018a; Xinyi and Chen, 2019), and find that GFN can match or exceed the best accuracies produced by recently proposed GNNs, at a fraction of the computation cost. GLN performs much poorly than both GFN and recent GNNs. This result casts doubts on the necessity of non-linear graph filtering, and suggests that the existing GNNs may not have learned more sophisticated graph functions than linear neighbor aggregation on these benchmarks. Furthermore, we find non-linear set function plays an important role, as its linearization can hurt performance significantly.

## 2 PRELIMINARIES

**Graph classification problem.** We use $G = (V, E) \in \mathcal{G}$ to denote a graph, where $V$ is a set of vertices/nodes, and $E$ is a set of edges. We further denote an attributed graph as $G_X = (G, X) \in \mathcal{G}_X$, where $X \in \mathbb{R}^{n \times d}$ are node attributes with $n = |V|$. It is assumed that each attributed graph is associated with some label $y \in \mathcal{Y}$, where $\mathcal{Y}$ is a set of pre-defined categories. The goal in graph classification problem is to learn a mapping function $f : \mathcal{G}_X \to \mathcal{Y}$, such that we can predict the target class for unseen graphs accurately. Many real world problems can be formulated as graph classification problems, such as social and biological graph classification Yanardag and Vishwanathan (2015); Kipf and Welling (2016).

**Graph neural networks.** Graph Neural Networks (GNNs) define functions on the space of attributed graph $\mathcal{G}_X$. Typically, the graph function, GNN$(G, X)$, learns a multiple-step transformation of the original attributes/signals for final node level or graph level prediction. In each of the step $t$, a new node presentation, $h_v^{(t)}$ is learned. Initially, $h_v^{(1)}$ is initialized with the node attribute vector, and during each subsequent step, a *neighbor aggregation function* is applied to generate the new node representation. More specifically, common neighbor aggregation functions for the $v$-th node take the following form:

$$h_v^{(t)} = f\left( h_v^{(t-1)}, \left\{ h_u^{(t-1)} | u \in \mathcal{N}(v) \right\} \right), \tag{1}$$

where $\mathcal{N}(v)$ is a set of neighboring nodes of node $v$. To instantiate this neighbor aggregation function, (Kipf and Welling, 2016) proposes the Graph Convolutional Network (GCN) aggregation scheme as follows.

$$h_v^{(t+1)} = \sigma\left( \sum_{u \in \mathcal{N}(v)} \tilde{A}_{uv} (W^{(t)})^T h_u^{(t)} \right), \tag{2}$$

where $W^{(t)} \in \mathbb{R}^{d \times d'}$ is the learnable transformation weight, $\tilde{A} = \tilde{D}^{-1/2}(A + \epsilon I)\tilde{D}^{-1/2}$ is the normalized adjacency matrix with $\epsilon$ as a constant ($\epsilon = 1$ in Kipf and Welling (2016)) and $\tilde{D}_{ii} = \sum_j A_{ij} + \epsilon$. $\sigma(\cdot)$ is a non-linear activation function, such as ReLU. This transformation can also be written as $H^{(t+1)} = \sigma(\tilde{A}H^{(t)}W^{(t)})$, where $H^{(t)} \in \mathbb{R}^{n \times d}$ are the hidden states of all nodes at $t$-th step.

More sophisticated neighbor aggregation schemes are also proposed, such as GraphSAGE (Hamilton et al., 2017) which allows pooling and recurrent aggregation over neighboring nodes. Most recently, in Graph Isomorphism Network (GIN) (Xu et al., 2019), a more powerful aggregation function is

proposed as follows.

$$h_v^{(t)} = \text{MLP}^{(t)} \left( \left( 1 + \epsilon^{(t)} \right) h_v^{(t-1)} + \sum_{u \in \mathcal{N}(v)} h_u^{(t-1)} \right), \tag{3}$$

where MLP abbreviates for multi-layer perceptrons and $\epsilon^{(t)}$ can either be zero or a learnable parameter.

Finally, in order to generate graph level representation $h_G$, a *readout function* is used, which generally takes the following form:

$$h_G = g \left( \left\{ h_v^{(T)} | v \in G \right\} \right). \tag{4}$$

This can be instantiated by a global sum pooling, i.e. $h_G = \sum_{v=1}^n h_v^{(T)}$ followed by fully connected layers to generate the categorical or numerical output.

## 3 APPROACH

### 3.1 GRAPH FEATURE NETWORK

Motivated by the question that, with a powerful graph readout function, whether wwe can simplify the sophisticated multi-step neighbor aggregation functions (such as Eq. 2 and 3). Therefore we propose Graph Feature Network (GFN): a neural set function defined on a set of graph augmented features.

**Graph augmented features.** In GFN, we replace the sophisticated neighbor aggregation functions (such as Eq. 2 and 3) with graph augmented features based on $G_X$. Here we consider two categories as follows: 1) graph structural/topological features, which are related to the intrinsic graph structure, such as node degrees, or node centrality scores[1], but do not rely on node attributes; 2) graph propagated features, which leverage the graph as a medium to propagate node attributes. The graph augmented features $X^G$ can be seen as the output of a feature extraction function defined on the attributed graph, i.e. $X^G = \gamma(G, X)$, and Eq. 5 below gives a specific form, which combine node degree features and multi-scale graph propagated features as follows:

$$X^G = \gamma(G, X) = \left[ \boldsymbol{d}, X, \tilde{A}^1 X, \tilde{A}^2 X, \cdots, \tilde{A}^K X \right], \tag{5}$$

where $\boldsymbol{d} \in \mathbb{R}^{n \times 1}$ is the degree vector for all nodes, and $\tilde{A}$ is again the normalized adjacency matrix ($\tilde{A} = \tilde{D}^{-1/2}(A + \epsilon I)\tilde{D}^{-1/2}$), but other designs of propagation operator are possible (Klicpera et al., 2019). Features separated by comma are concatenated to form $X^G$.

**Neural set function.** To build a powerful graph readout function based on graph augmented features $X^G$, we use a neural set function. The neural set function discards the graph structures and learns purely based on the set of augmented node features. Motivated by the general form of a permutation-invariant set function shown in Zaheer et al. (2017), we define our neural set function for GFN as follows:

$$\text{GFN}(G, X) = \rho \left( \sum_{v \in \mathcal{V}} \phi \left( X_v^G \right) \right). \tag{6}$$

Both $\phi(\cdot)$ and $\rho(\cdot)$ are parameterized by neural networks. Concretely, we parameterize the function $\phi(\cdot)$ as a multi-layer perceptron (MLP), i.e. $\phi(x) = \sigma(\sigma(\cdots \sigma(x^T W^{(1)}) \cdots) W^{(T)})$. Note that a single layer of $\phi(\cdot)$ resembles a graph convolution layer $H^{(t+1)} = \sigma(\tilde{A} H^{(t)} W^{(t)})$ with the normalized adjacency matrix $\tilde{A}$ replaced by identity matrix $I$ (a.k.a. $1 \times 1$ convolution). As for the function $\rho(\cdot)$, we parameterize it with another MLP (i.e. fully connected layers in this case).

---

[1]We only use node degree in this work as it is very efficient to calculate during both training and inference.

**Computation efficiency.** GFN provides a way to approximate GNN with less computation overheads, especially during the training process. Since the graph augmented features can be pre-computed before training starts, the graph structures are not involved in the iterative training process. This brings the following advantages. First, since there is no neighbor aggregation step in GFN, it reduces computational complexity. To see this, one can compare a single layer feature transformation function in GFN, i.e. $\sigma(HW)$, against the neighbor aggregation function in GCN, i.e. $\sigma(\tilde{A}HW)$. Secondly, since graph augmented features of different scales are readily available from the input layer, GFN can leverage them much earlier, thus may require fewer transformation layers. Lastly, it also eases the implementation related overhead, since the neighbor aggregation operation in graphs are typically implemented by sparse matrix operations.

**Graph Linear Network.** When we use a linear set function instead of the generic one used in Eq. 6, we arrive at graph linear network, which can be expressed as follows.

$$\text{GLN}(G, X) = \sigma\left(W \sum_{v \in \mathcal{V}} \left(X_v^G\right)\right). \tag{7}$$

Where $W$ is a weight matrix, and $\sigma(\cdot)$ is softmax function produce class probability.

## 3.2 FROM GNN TO GFN AND GLN: A DISSECTION OF GNNs

To better understand GNNs on graph classification, we propose a formal dissection/decomposition of GNNs into two parts/stages: the graph filtering part and the set function part. As we shall see shortly, the simplification of the graph filtering part allows us to derive GFN from GNN, and also be able to assess the importance of the two GNN parts separately.

To make concepts more clear, we first give formal definitions of the two GNN parts in the dissection.

**Definition 1.** (Graph filtering) A graph filtering function, $Y = \mathcal{F}_G(X)$, performs a transformation of input signals based on the graph $G$, which takes a set of signals $X \in \mathbb{R}^{n \times d}$ and outputs another set of filtered signals $Y \in \mathbb{R}^{m \times d'}$.

Graph filtering in most existing GNNs consists of multi-step neighbor aggregation operations, i.e. multiple steps of Eq. 1. For example, in GCN Kipf and Welling (2016), the multi-step neighbor aggregation can be expressed as $H^{(T)} = \sigma(A\sigma(...\sigma(AXW^{(1)})...)W^{(T)})$.

**Definition 2.** (Set function) A set function, $y = \mathcal{T}(Y)$, takes a set of vectors $Y \in \mathbb{R}^{m \times d'}$ where their order does not matter, and outputs a task specific prediction $y \in \mathcal{Y}$.

The graph readout function in Eq. 4 is a set function, which enables the graph level prediction that is permutation invariant w.r.t. nodes in the graph. Although a typical readout function is simply a global pooling (Xu et al., 2019), the set function can be as complicated as Eq. 6.

**Claim 1.** *A GNN that is a mapping of $\mathcal{G}_X \to \mathcal{Y}$ can be decomposed into a graph filtering function followed by a set function, i.e. $GNN(G, X) = \mathcal{T} \circ \mathcal{F}_G(X)$.*

This claim is obvious for the neighbor aggregation framework defined by Eq. 1 and 4, where most existing GNN variants such as GCN, GraphSAGE and GIN follow. This claim is also general, even for unforeseen GNN variants that do not explicitly follow this framework [2].

We aim to assess the importance of two GNN parts separately. However, it is worth pointing out that the above decomposition is not unique in general, and the functionality of the two parts can overlap: if the graph filtering part has fully transformed graph features, then a simple set function may be used for prediction. This makes it challenging to answer the question: do we need a sophisticated graph filtering part for a particular task or dataset, especially when a powerful set function is used? To better disentangle these two parts and study their importance more independently, similar to Wu et al. (2019), we propose to simplify the graph filtering part by linearizing it.

**Definition 3.** (Linear graph filtering) We say a graph filtering function $\mathcal{F}_G(X)$ is linear w.r.t. $X$ iff it can be expressed as $\mathcal{F}_G(X) = \Gamma(G, X)\boldsymbol{\theta}$, where $\Gamma(G, X)$ is a linear map of $X$, and $\boldsymbol{\theta}$ is the only learnable parameter.

---

[2]We can absorb the set function $\mathcal{T}$ into $\mathcal{F}_G$. That is, let the output $Y = \mathcal{F}_G(\cdot)$ be final logits for pre-defined classes and set $\mathcal{T}(\cdot)$ to softmax function with zero temperature, i.e. $\exp(x/\tau)/Z$ with $\tau \to 0$

Intuitively, one can construct a linear graph filtering by removing the non-linear operations from graph filtering part in existing GNNs, such as non-linear activation function $\sigma(\cdot)$ in Eq. 2 or 3. By doing so, the graph filtering becomes linear w.r.t. X, thus multi-layer weights collapse into a single linear transformation, described by $\boldsymbol{\theta}$. More concretely, let us consider a linearized GCN Kipf and Welling (2016), its $K$-th layer can be written as $H^{(K)} = \tilde{A}^K X(\Pi_{k=1}^K W^{(k)})$, and we can rewrite the weights with $\boldsymbol{\theta} = \Pi_{k=1}^K W^{(k)}$.

The linearization of graph filtering part enables us to disentangle graph filtering and the set function more thoroughly: the graph filtering part mainly constructs graph augmented features (by setting $\gamma(G, X) = \Gamma(G, X)$), and the set function learns to compose them for the graph-level prediction. This leads to the proposed GFN. In other words, GNNs with a linear graph filtering part can be expressed as GFN with appropriate graph augmented features. This is shown more formally in the following proposition 1.

**Proposition 1.** *Let $GNN^{lin}(G, X)$ be a mapping of $\mathcal{G}_X \to \mathcal{Y}$ that has a linear graph filtering part, i.e. $\mathcal{F}_G(X) = \Gamma(G, X)\boldsymbol{\theta}$, then we have $GNN^{lin}(G, X) = GFN(G, X)$, where $\gamma(G, X) = \Gamma(G, X)$.*

The proof can be found in the appendix. Noted that a GNN with a linear graph filtering can be seen as a GFN, but the reverse may not be true. General GFN can have non-linear graph filtering, e.g. when the feature extraction function $\gamma(G, X)$ is not a linear map of $X$ (Eq. 5 is a linear map of $X$).

**Why GFN?** GFN can also help us understand the functions that GNNs learned on current benchmarks. First, by comparing GNN with linear graph filtering (i.e. GFN) against standard GNN with non-linear graph filtering, we can assess the importance of non-linear graph filtering part. Secondly, by comparing GFN with linear set function (i.e. GLN) against GFN with non-linear set function, we can assess the importance of non-linear set function.

Beyond as a tool to study GNN parts, GFN is also more efficient than GNN counterpart, which makes it a fast approximation. Furthermore, GFNs can be a very powerful framework without restriction on the feature extraction function $\gamma(G, X)$ and the exact forms of the set function. The potential expressiveness of a GFN is demonstrated by the following proposition.

**Proposition 2.** *For any GNN $\mathcal{F}$ defined in $\mathcal{G}_X$, there exists a graph to set mapping $\mathcal{M} : \mathcal{G} \to \mathcal{S}$ where $\mathcal{S}$ is a set space, and a set function $\mathcal{T}$ that approximates $\mathcal{F}$ to arbitrary precision, i.e. $\forall G \in \mathcal{G}_X, F(G) \approx \mathcal{T}(\mathcal{M}(G))$.*

The proof is provided in the appendix. We want to provide an intuitive interpretation here. There exists some way(s) that we can encode any graph into a set, and learn a generic set function on it. As long as the set contains the graph information, a powerful set function can learn to integrate it in a flexible way. So a well constructed GFN can be as powerful as, if not more powerful than, the most powerful GNNs. This shows the potential of the GFN framework in modeling arbitrary graph data.

## 4 EXPERIMENTS

### 4.1 DATASETS AND SETTINGS

**Datasets.** The main datasets we consider are commonly used graph classification benchmarks (Yanardag and Vishwanathan, 2015; Xinyi and Chen, 2019; Xu et al., 2019). The graphs in the collection can be categorized into two categories: (1) biological graphs, including MUTAG, NCI1, PROTEINS, D&D, ENZYMES; and (2) social graphs, including COLLAB, IMDB-Binary (IMDB-B), IMDB-Multi (IMDB-M), Reddit-Multi-5K (RE-M5K), Reddit-Multi-12K (RE-M12K). It is worth noting that the social graphs have no node attributes, while the biological graphs come with categorical node attributes. The detailed statistics can be found in the appendix.

**Baselines.** We compare with two families of baselines. The first family of baselines are kernel-based, namely the Weisfeiler-Lehman subtree kernel (WL) (Shervashidze et al., 2011), Deep Graph Kernel (DGK) (Yanardag and Vishwanathan, 2015) and AWE (Ivanov and Burnaev, 2018) that incorporate kernel-based methods with learning-based approach to learn embeddings. The second family of baselines are GNN-based models, which include recently proposed PATCHY-SAN (PSCN) (Niepert

Table 1: Test accuracies (%) for biological graphs. The best results per dataset and in average are highlighted. - means the results are not available for a particular dataset.

| Algorithm | MUTAG | NCI1 | PROTEINS | D&D | ENZYMES | Average |
|---|---|---|---|---|---|---|
| WL | 82.05±0.36 | 82.19±0.18 | 74.68±0.49 | **79.78±0.36** | 52.22±1.26 | 74.18 |
| AWE | 87.87±9.76 | - | - | 71.51±4.02 | 35.77±5.93 | - |
| DGK | 87.44±2.72 | 80.31±0.46 | 75.68±0.54 | 73.50±1.01 | 53.43±0.91 | 74.07 |
| PSCN | 88.95±4.37 | 76.34±1.68 | 75.00±2.51 | 76.27±2.64 | - | - |
| DGCNN | 85.83±1.66 | 74.44±0.47 | 75.54±0.94 | 79.37±0.94 | 51.00±7.29 | 73.24 |
| CapsGNN | 86.67±6.88 | 78.35±1.55 | 76.28±3.63 | 75.38±4.17 | 54.67±5.67 | 74.27 |
| GIN | 89.40±5.60 | 82.70±1.70 | 76.20±2.80 | - | - | - |
| GCN | 87.20±5.11 | **83.65±1.69** | 75.65±3.24 | 79.12±3.07 | 66.50±6.91 | 78.42 |
| GLN | 82.85±12.15 | 68.61±2.31 | 75.65±4.43 | 76.75±5.00 | 43.83±5.16 | 69.54 |
| GFN | **90.84±7.22** | 82.77±1.49 | 76.46±4.06 | 78.78±3.49 | **70.17±5.58** | **79.80** |
| GFN-light | 89.89±7.14 | 81.43±1.65 | **77.44±3.77** | 78.62±5.43 | 69.50±7.37 | 79.38 |

Table 2: Test accuracies (%) for social graphs. The best results per dataset and in average are highlighted. - means the results are not available for a particular dataset.

| Algorithm | COLLAB | IMDB-B | IMDB-M | RE-M5K | RE-M12K | Average |
|---|---|---|---|---|---|---|
| WL | 79.02±1.77 | 73.40±4.63 | 49.33±4.75 | 49.44±2.36 | 38.18±1.30 | 57.87 |
| AWE | 73.93±1.94 | 74.45±5.83 | 51.54±3.61 | 50.46±1.91 | 39.20±2.09 | 57.92 |
| DGK | 73.09±0.25 | 66.96±0.56 | 44.55±0.52 | 41.27±0.18 | 32.22±0.10 | 51.62 |
| PSCN | 72.60±2.15 | 71.00±2.29 | 45.23±2.84 | 49.10±0.70 | 41.32±0.42 | 55.85 |
| DGCNN | 73.76±0.49 | 70.03±0.86 | 47.83±0.85 | 48.70±4.54 | - | - |
| CapsGNN | 79.62±0.91 | 73.10±4.83 | 50.27±2.65 | 52.88±1.48 | 46.62±1.90 | 60.50 |
| GIN | 80.20±1.90 | **75.10±5.10** | **52.30±2.80** | 57.50±1.50 | - | - |
| GCN | **81.72±1.64** | 73.30±5.29 | 51.20±5.13 | 56.81±2.37 | 49.31±1.44 | 62.47 |
| GLN | 75.72±2.51 | 73.10±3.18 | 50.40±5.61 | 52.97±2.58 | 39.84±0.95 | 58.41 |
| GFN | 81.50±2.42 | 73.00±4.35 | 51.80±5.16 | **57.59±2.40** | 49.43±1.36 | **62.66** |
| GFN-light | 81.34±1.73 | 73.00±4.29 | 51.20±5.71 | 57.11±1.46 | **49.75±1.19** | 62.48 |

et al., 2016), Deep Graph CNN (DGCNN) (Zhang et al., 2018a), CapsGNN (Xinyi and Chen, 2019) and GIN (Xu et al., 2019).

For the above baselines, we use their accuracies reported in the original papers, following the same evaluation setting as in (Xu et al., 2019). Architecture and hyper-parameters can make a difference, so to enable a better controlled comparison between GFN and GNN, we also implement Graph Convolutional Networks (GCN) from (Kipf and Welling, 2016). More specifically, our GCN model contains a dense feature transformation layer, i.e. $H^{(2)} = \sigma(XW^{(1)})$, followed by three GCN layers, i.e. $H^{(t+1)} = \sigma(\tilde{A}H^{(t)}W^{(t)})$. We also vary the number of GCN layers in our ablation study. To enable graph level prediction, we add a global sum pooling, followed by two fully-connected layers that produce categorical probability over pre-defined categories.

**Model configurations.** For the proposed GFN, we *mirror* our GCN model configuration to allow direct comparison. Therefore, we use the same architecture, parameterization and training setup, but replace the GCN layer with feature transformation layers (totaling four such layers). Converting GCN layer to feature transformation layer is equivalent to setting $A = I$ in in GCN layers. We also construct a faster GFN, namely "GFN-light", that contains only a single feature transformation layer, which can further reduce the training time while maintaining similar performance.

For both our GCN and GFN, we utilize ReLU activation and batch normalization (Ioffe and Szegedy, 2015), and fix the hidden dimensionality to 128. No regularization is applied. Furthermore we use batch size of 128, a fixed learning rate of 0.001, and the Adam optimizer (Kingma and Ba, 2014). GLN follows the same setting as GFN, but contains no feature transform layer. It only has the global sum pooling of graph features followed by a single fully connected layer. To compare with existing

work, we follow (Xinyi and Chen, 2019; Xu et al., 2019) and perform 10-fold cross validation. We run the model for 100 epochs, and select the epoch in the same way as Xu et al. (2019), i.e., a single epoch with the best cross-validation accuracy averaged over the 10 folds is selected. We report the average and standard deviation of test accuracies at the selected epoch over 10 folds.

In terms of input node features for GFN and GLN, by default, we use both degree and multi-scale propagated features (up to $K = 3$), that is $[\boldsymbol{d}, X, \tilde{A}^1 X, \tilde{A}^2 X, \tilde{A}^3 X]$. We turn discrete features into one-hot vectors, and also discretize degree features into one-hot vectors, as suggested in Fey and Lenssen (2019). We set $X = \vec{1}$ for the social graphs we consider as there are no node attributes. By default, we also augment node features in our GCN with an extra node degree feature (to counter that the normalized adjacency matrix may lose the degree information). Other graph augmented features are also studied for GCN (which has minor effects). All experiments are run on Nvidia GTX 1080 Ti GPU.

## 4.2 PERFORMANCE COMPARISON BETWEEN GLN, GFN AND EXISTING GNN VARIANTS

Table 1 and 2 show the results of different methods in both biological and social datasets. It is worth noting that in both datasets, GFN achieves similar performances with our GCN, and match or exceed existing state-of-the-art results on multiple datasets, while GLN performs worse in most of the datasets. This result suggests the importance of non-linear set function, while casting doubt on the necessity of non-linear graph filtering for these benchmarks.

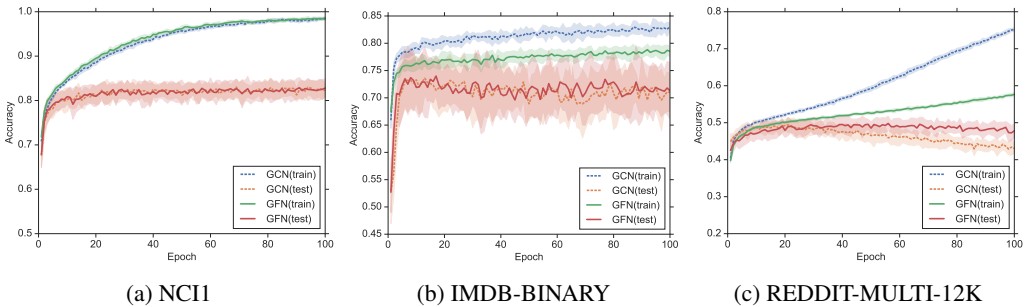

|     (a) NCI1     | (b) IMDB-BINARY | (c) REDDIT-MULTI-12K |
| --- | --- | --- |

Figure 1: Training and test performance versus training epoch.

Figure 1 shows training/test curves for both GCN and GFN. We observe that GCN usually perform better than GFN during the training, but their test performances are mostly similar (sometimes GFN is better as training continues). This concludes that GFN works well not because it is easier to optimize.

## 4.3 TRAINING TIME COMPARISONS BETWEEN GFNS AND GCNS

Since GFN's performance is on par with GCN's, we further compare the training time of our GCN and the proposed GFNs. Figure 2 shows that a significant speedup (from $1.4\times$ to $6.7\times$ as fast) by utilizing GFN compared to GCN, especially for datasets with denser edges such as the COLLAB dataset. Also since our GFN can work with fewer transformation layers, GFN-light can achieve better speedup by reducing the number of transformation layers. Note that our GCN is already very efficient as it is built on a highly optimized framework Fey and Lenssen (2019).

## 4.4 ABLATIONS

**Node features.** To better understand the impact of features, we test both models with different input node features. Table 3 shows that 1) graph features are very important for both GFN and GCN, 2) the node degree feature is surprisingly important, and multi-scale features can further improve on that, and 3) even with multi-scale features, GCN still performs similarly to GFN, which further suggests that linear graph filtering is enough. More detailed results (per dataset) can be found in the appendix.

**Architecture depth.** We vary the number of convolutional layers (with two FC-layers after sum pooling kept the same). Table 4 shows that 1) GCN benefits from multiple grpah convolutional layers with a significant diminishing return, 2) GFN with single feature transformation layer works pretty

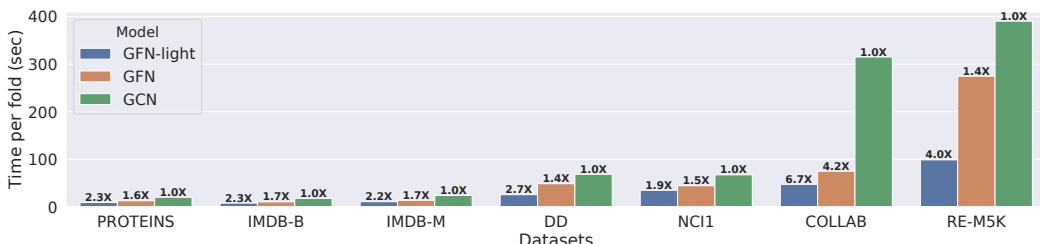

Figure 2: Training time comparisons. The annotation, e.g. $1.0\times$, denotes speedup compared to GCN.

Table 3: Accuracies (%) under various augmented features. Averaged results over multiple datasets are shown here. $A^{1,2,3}X$ is abbreviated for $A^1X, A^2X, A^3X$, and default node feature $X$ is always used (if available) but not displayed to reduce clutter. Best results per row/block are highlighted.

| Graphs | Model | None | $d$ | $A^1X$ | $A^{1,2}X$ | $A^{1,2,3}X$ | $d, A^1X$ | $d, A^{1,2}X$ | $d, A^{1,2,3}X$ |
|--------|-------|------|-----|--------|-----------|-------------|-----------|--------------|----------------|
| Bio. | GCN | **78.52** | 78.51 | 78.23 | 78.24 | **78.68** | 79.10 | 79.26 | **79.69** |
|      | GFN | **76.27** | 77.84 | 78.78 | 79.09 | **79.17** | 78.71 | **79.21** | 79.13 |
| Soical | GCN | **34.02** | **62.35** | 59.20 | 60.39 | 60.28 | 62.45 | 62.71 | **62.77** |
|       | GFN | **30.45** | **60.79** | 58.04 | 59.83 | 60.09 | 62.47 | **62.63** | 62.60 |

Table 4: Accuracies (%) under different number of Conv. layers.

|  |  | 1 | 2 | 3 | 4 | 5 |
|--------|-------|------|------|------|------|------|
| Bio. | GCN | 77.17 | **79.38** | 78.86 | 78.75 | 78.21 |
|      | GFN | 79.59 | 79.77 | **79.78** | 78.99 | 78.14 |
| Soical | GCN | 60.69 | 62.12 | 62.37 | **62.70** | 62.46 |
|       | GFN | 62.70 | **62.88** | 62.81 | 62.80 | 62.60 |

well already, likely due to the availability of multi-scale input node features, which otherwise require multiple GCN layers to obtain.

**Visualization.** We also provide visualization of random and misclassified samples from the tested graph datasets in the appendix J. We could not clearly distinguish graphs from different classes easily based on their appearance, suggesting that both GFN and GCN are capturing underlying non-trivial features.

## 5 DISCUSSION

In this work, we conduct a dissection of GNNs on common graph classification benchmarks. We first decompose GNNs into two parts, and linearize the graph filtering part resulting GFN. We then further linearize the set function of GFN resulting GLN. In our extensive experiments, we find GFN can match or exceed the best results by recently proposed GNNs, with a fraction of computation cost. The linearization of graph filtering (i.e. GFN) has little impact on performance, while linearization of both graph filtering and set function (i.e. GLN) leads to worse performance.

Since GCN usually achieve better training accuracies while not better test accuracies, we conjecture that the linear graph filtering may be a good inductive bias for tested datasets, though this is speculative and requires more future investigations. Another possibility is that complexity of current graph classification benchmarks is limited, so that linear graph filtering is enough, thus moving to datasets or problems with information that is more structurally complicated could require sophisticated non-linear graph filtering.

We also observe the potential of GFN, which leverages a generic set function to model graphs. In the future, we would like to build upon the powerful general GFN framework for structured data.

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

## A  COMPARISONS OF DIFFERENT LINEARIZATIONS

Table 5: Comparisons of different linearizations.

| Method | Graph filtering | Set function | Efficiency | Performance |
|--------|-----------------|--------------|------------|-------------|
| GLN | Linear | Linear | High | Low |
| GFN$^{lin}$ | Linear | Non-linear | High | High |
| GCN | Non-linear | Linear/Non-linear | Low | High |

Table 5 summarizes the comparisons between GCN and its linearized variants. The efficiency and performance are concluded from our experiments on graph classification benchmarks. Noted that a GNN with a linear graph filtering can be seen as a GFN, but the reverse may not be true. General GFN can have non-linear graph filtering, e.g. when the feature extraction function $\gamma(G, X)$ is not a linear map of $X$. Thus we use GFN$^{lin}$ in Table 5 to denote such subtle difference.

## B  PROOFS

Here we provide the proof for Proposition 1.

*Proof.* According to claim 1 and definition 3, a GNN$(G, X)$ with a linear graph filtering part, denoted by GNN$^{lin}(G, X)$, can be written as follows.

$$\text{GNN}^{lin}(G, X) = \mathcal{T} \circ \mathcal{F}_G(X) = \mathcal{T}(\Gamma(G, X)\boldsymbol{\theta}) = \mathcal{T}'(\Gamma(G, X)),$$

where $\boldsymbol{\theta}$ is absorbed into the set function $\mathcal{T}'(\cdot)$. According to GFN's definition in Eq. 6 and general set function result from Zaheer et al. (2017), we have

$$\text{GFN}(G, X) = \mathcal{T}''(X^G) = \mathcal{T}''(\gamma(G, X)).$$

By setting $\gamma(G, X) = \Gamma(G, X)$, we arrive at GNN$^{lin}(G, X) = $ GFN$(G, X)$.  □

Here we provide the proof for Proposition 2.

*Proof.* We show the existence of the mapping $\mathcal{T}$ by constructing it as follows. First, we assign a unique ID to each of the node, then we add its ID and its neighbors' IDs in the end of node features. If there are edges with features, we also treat them as nodes and apply the same above procedure. This procedure results in a set of nodes with features that preserve the same original information (since we can reconstruct the original graph).

We now show the existence of a set function that can mimic any graph functions operated on $\mathcal{G}$, again, by constructing a specific one. Since the set of nodes preserve the whole graph information, the set function can first reconstruct the graph by decoding the node's feature vectors. At every computation step, the set function find neighbors of each node in the set, and compute the aggregation function in exactly the same way as the graph function would do with the neighbors of a node. This procedure is repeated until the graph function produces its output.

Hence, the above constructed example proves the existence of $\mathcal{M}$ and a set function $\mathcal{T}$ such that $\forall G \in \mathcal{G}_X, \mathcal{F}(G) \approx \mathcal{T}(\mathcal{M}(G))$. We also note that the specially constructed examples above are feasible but likely not optimal. A better solution is to have a set function that learns to adaptively leverage the graph structure as well as node attributes.  □

## C  DETAILED STATISTICS OF DATASETS

Detailed statistics of the biological and social graph datasets are listed in Table 6 and 7, respectively.

Table 6: Data statistics of Biological dataset

| Dataset | MUTAG | NCI1 | PROTEINS | D&D | ENZYMES |
|---|---|---|---|---|---|
| # graphs | 188 | 4110 | 1113 | 1178 | 600 |
| # classes | 2 | 2 | 2 | 2 | 6 |
| # features | 7 | 37 | 3 | 82 | 3 |
| Avg # nodes | 17.93 | 29.87 | 39.06 | 284.32 | 32.63 |
| Avg # edges | 19.79 | 32.30 | 72.82 | 715.66 | 62.14 |

Table 7: Data statistics of Social dataset

| Dataset | COLLAB | IMDB-B | IMDB-M | RE-M5K | RE-12K |
|---|---|---|---|---|---|
| # graphs | 5000 | 1000 | 1500 | 4999 | 11929 |
| # classes | 3 | 2 | 3 | 5 | 11 |
| # features | 1 | 1 | 1 | 1 | 1 |
| Avg # nodes | 74.49 | 19.77 | 13.00 | 508.52 | 391.41 |
| Avg # edges | 2457.78 | 96.53 | 65.94 | 594.87 | 456.89 |

## D    EXPERIMENTS ON GRAPH CONSTRUCTED FROM IMAGES (MNIST)

In addition to the common graph benchmarks, we also consider image classification on MNIST where pixels are treated as nodes and eight nearest neighbors in the grid, with an extra self-loop, are used to construct the graph.

For MNIST, we train and evaluate on the given train/test split. Additionally, since MNIST benefits more from deeper GCN layers, we parameterize our GCN model using a residual network (He et al., 2016) with multiple GCN blocks, the number of blocks are kept the same for GCN and GFN, and varied according to the size of total receptive field. GFN utilizes the same multi-scale features as in Eq. 5.

We report the accuracies under different total receptive field sizes (i.e. the number of hops a pixel could condition its computation on). Results in Table 8 show that, in all three different receptive field sizes, GCN with non-linear neighbor aggregation outperforms GFN with linear graph propagated features. This indicates that non-linear graph filtering is essential for performing well in this dataset. Note that our results are not directly comparable to traditional CNN's, as our GNN does not distinguish the neighbor pixel direction in its parameterization, and a global sum pooling of pixels does not leverage spatial information. For context, when using coordinates as features both GCN and GFN achieve nearly 99% accuracy.

Table 8: Test accuracies (%) on MNIST graphs.

| Receptive size | GCN | GFN |
|---|---|---|
| 3 | **91.47** | 87.73 |
| 5 | **95.16** | 91.83 |
| 7 | **96.14** | 92.68 |

This test on graphs constructed from image dataset (MNIST), the observation that similarly configured GCN outperforms GFN by a large margin, indicates the importance of non-linear graph filtering for this type of graph dataset.

## E    DETAILED PERFORMANCES WITH DIFFERENT FEATURES

Table 9 show the performances under different graph features for GNNs and GFNs. It is evident that both model benefit significantly from graph features, especially GFNs.

## F    DETAILED PERFORMANCES WITH DIFFERENT ARCHITECTURE DEPTHS

Table 10 shows performance per datasets under different number of layers.

Table 9: Accuracies (%) under various augmented features. $A^{1..3}X$ is abbreviated for $A^1X, A^2X, A^3X$, and default node feature $X$ is always used (if available) but not displayed to reduce clutter.

| Dataset | Model | None | $d$ | $A^1X$ | $A^{1,2}X$ | $A^{1..3}X$ | $d, A^1X$ | $d, A^{1,2}X$ | $d, A^{1..3}X$ |
|---------|-------|------|-----|--------|-----------|-------------|-----------|---------------|----------------|
| MUTAG | GCN | 83.48 | 87.09 | 83.35 | 83.43 | 85.56 | 87.18 | 87.62 | 88.73 |
|        | GFN | 82.21 | 89.31 | 87.59 | 87.17 | 86.62 | 89.42 | 89.28 | 88.26 |
| NCI1 | GCN | 80.15 | 83.24 | 82.62 | 83.11 | 82.60 | 83.38 | 83.63 | 83.50 |
|      | GFN | 70.83 | 75.50 | 80.95 | 82.80 | 83.50 | 81.92 | 82.41 | 82.84 |
| PROTEINS | GCN | 74.49 | 76.28 | 74.48 | 75.47 | 76.54 | 77.09 | 76.91 | 77.45 |
|          | GFN | 74.93 | 76.63 | 76.01 | 75.74 | 76.64 | 76.37 | 76.46 | 77.09 |
| DD | GCN | 79.29 | 78.78 | 78.70 | 77.67 | 78.18 | 78.35 | 78.79 | 79.12 |
|    | GFN | 78.70 | 77.77 | 77.85 | 77.43 | 78.28 | 77.34 | 76.92 | 78.11 |
| ENZYMES | GCN | 75.17 | 67.17 | 72.00 | 71.50 | 70.50 | 69.50 | 69.33 | 69.67 |
|         | GFN | 74.67 | 70.00 | 71.50 | 72.33 | 70.83 | 68.50 | 71.00 | 69.33 |
| COLLAB | GCN | 39.69 | 82.14 | 76.62 | 76.98 | 77.22 | 82.14 | 82.24 | 82.20 |
|        | GFN | 31.57 | 80.36 | 76.40 | 77.08 | 77.04 | 81.28 | 81.62 | 81.26 |
| IMDB-B | GCN | 51.00 | 73.00 | 70.30 | 71.10 | 72.20 | 73.50 | 73.80 | 73.70 |
|        | GFN | 50.00 | 73.30 | 72.30 | 71.30 | 71.70 | 74.40 | 73.20 | 73.90 |
| IMDB-M | GCN | 35.00 | 50.33 | 45.53 | 46.33 | 45.73 | 50.20 | 50.73 | 51.00 |
|        | GFN | 33.33 | 51.20 | 46.80 | 46.67 | 46.47 | 51.93 | 51.93 | 51.73 |
| RE-M5K | GCN | 28.48 | 56.99 | 54.97 | 57.43 | 56.55 | 56.67 | 56.75 | 57.01 |
|        | GFN | 20.00 | 54.23 | 51.11 | 55.85 | 56.35 | 56.45 | 57.01 | 56.71 |
| RE-M12K | GCN | 15.93 | 49.28 | 48.58 | 50.11 | 49.71 | 49.73 | 50.03 | 49.92 |
|         | GFN | 17.33 | 44.86 | 43.61 | 48.25 | 48.87 | 48.31 | 49.37 | 49.39 |

Table 10: Accuracies (%) under different number of Conv. layers. Flat denotes the collapsed GFN into a linear model (i.e. linearizing the set function).

| Dataset | Method | Flat | 1 | 2 | 3 | 4 | 5 |
|---------|--------|------|---|---|---|---|---|
| MUTAG | GCN | - | 88.32 | 90.89 | 87.65 | 88.31 | 87.68 |
|       | GFN | 82.85 | 90.34 | 89.39 | 88.18 | 87.59 | 87.18 |
| NCI1 | GCN | - | 75.62 | 81.41 | 83.04 | 82.94 | 83.31 |
|      | GFN | 68.61 | 81.77 | 83.09 | 82.85 | 82.80 | 83.09 |
| PROTEINS | GCN | - | 76.91 | 76.99 | 77.00 | 76.19 | 75.29 |
|          | GFN | 75.65 | 77.71 | 77.09 | 77.17 | 76.28 | 75.92 |
| DD | GCN | - | 77.34 | 77.93 | 78.95 | 79.46 | 78.77 |
|    | GFN | 76.75 | 78.44 | 78.78 | 79.04 | 78.45 | 76.32 |
| ENZYMES | GCN | - | 67.67 | 69.67 | 67.67 | 66.83 | 66.00 |
|         | GFN | 43.83 | 69.67 | 70.50 | 71.67 | 69.83 | 68.17 |
| COLLAB | GCN | - | 80.36 | 81.86 | 81.40 | 81.90 | 81.78 |
|        | GFN | 75.72 | 81.24 | 82.04 | 81.36 | 82.18 | 81.72 |
| IMDB-B | GCN | - | 72.60 | 72.30 | 73.30 | 73.80 | 73.40 |
|        | GFN | 73.10 | 73.50 | 73.30 | 74.00 | 73.90 | 73.60 |
| IMDB-M | GCN | - | 51.53 | 51.07 | 50.87 | 51.53 | 50.60 |
|        | GFN | 50.40 | 51.73 | 52.13 | 51.93 | 51.87 | 51.40 |
| RE-M5K | GCN | - | 54.05 | 56.49 | 56.83 | 56.73 | 56.89 |
|        | GFN | 52.97 | 57.45 | 57.13 | 57.21 | 56.61 | 57.03 |
| RE-M12K | GCN | - | 44.91 | 48.87 | 49.45 | 49.52 | 49.61 |
|         | GFN | 39.84 | 49.58 | 49.82 | 49.54 | 49.44 | 49.27 |

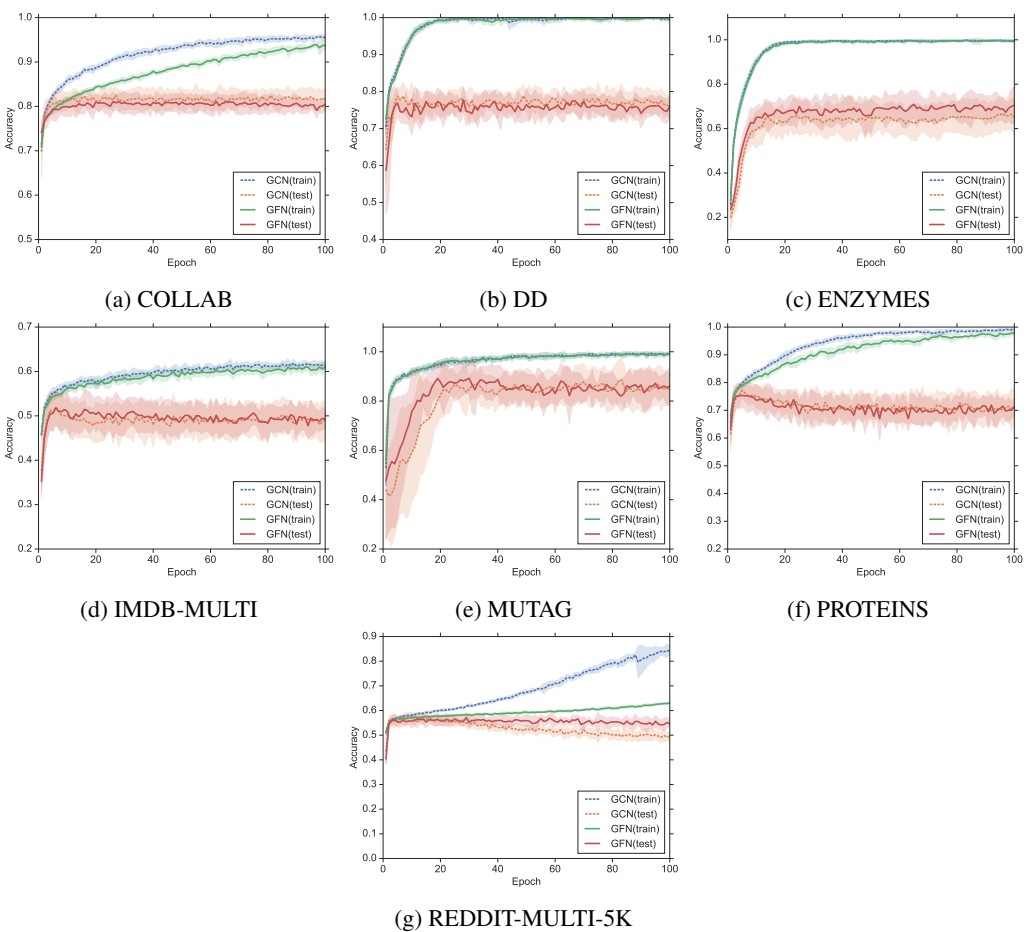

(a) COLLAB        (b) DD        (c) ENZYMES

(d) IMDB-MULTI        (e) MUTAG        (f) PROTEINS

(g) REDDIT-MULTI-5K

Figure 3: Training and test performance versus training epoch.

## G   MORE CURVES ON TRAINING / TEST PERFORMANCE VS EPOCH

Figure 3 shows more training/test curves for both GCN and GFN. The conclusion is consistent with main text that GFN works well not because it is easier to optimize.

## H   COMPARISONS TO RETGK AND GNTK

Here we further compare our results to two recent work, namely RetGK (Zhang et al., 2018b) and GNTK (Du et al., 2019). RetGK proposes a family of graph kernels based on return probabilities of random walks, with different instantiations: $RetGK_I$, $RetGK_{II}$, and $RetGK_{II}$(MC). In their experiments, node attribute are divided into three types: non-attribute, discrete attributes, and continues attributes, and we compare to their reported results. GNTK leverages the connection between infinitely wide networks and kernels to construct an infinitely wide GNN using graph kernels. We also compare to their reported results.

It is worth mentioning that these graph kernel based methods are typically quadratic in the number of graphs and nodes, which makes them hard to scale to large datasets. The proposed GFN has linear complexity and even faster than typical GNNs, which makes our method really scalable to larger datasets.

The results on biological and social graphs are shown in Table 11 and 12 respectively. We found that overall, despite the methodology differences, GFN still performs on par with these methods averaged over compared datasets (with performance differences on some datasets but they are mostly within one standard deviation).

Table 11: Test accuracies (%) for biological graphs. The best results per dataset and in average are highlighted. - means the results are not available for a particular dataset.

| Algorithm | MUTAG | NCI1 | PROTEINS | D&D | ENZYMES | **Average** |
|---|---|---|---|---|---|---|
| RetGK$_I$(Dis) | **90.3±1.1** | **84.5±0.2** | 75.8±0.6 | **81.6±0.3** | 60.4±0.8 | **78.52** |
| RetGK$_{II}$(Dis) | 90.1±1.0 | 83.5±0.2 | 75.2±0.3 | 81.0±0.5 | 59.1±1.1 | 77.78 |
| RetGK$_I$(Con) | - | - | 76.2±0.5 | - | 70.0±0.9 | - |
| RetGK$_{II}$(Con) | - | - | 75.9±0.4 | - | 70.7±0.9 | - |
| RetGK$_I$(Con&Dis) | - | - | **78.0±0.3** | - | **72.2±0.8** | - |
| RetGK$_{II}$(Con&Dis) | - | - | 77.3±0.5 | - | 70.6±0.7 | - |
| GNTK | 90.00±8.5 | 84.2±1.5 | 75.6±4.2 | - | - | - |
| GCN | 87.20±5.11 | **83.65±1.69** | 75.65±3.24 | **79.12±3.07** | 66.50±6.91 | 78.42 |
| GLN | 82.85±12.15 | 68.61±2.31 | 75.65±4.43 | 76.75±5.00 | 43.83±5.16 | 69.54 |
| GFN | **90.84±7.22** | 82.77±1.49 | 76.46±4.06 | 78.78±3.49 | **70.17±5.58** | **79.80** |
| GFN-light | 89.89±7.14 | 81.43±1.65 | **77.44±3.77** | 78.62±5.43 | 69.50±7.37 | 79.38 |

Table 12: Test accuracies (%) for social graphs. The best results per dataset and in average are highlighted. - means the results are not available for a particular dataset.

| Algorithm | COLLAB | IMDB-B | IMDB-M | RE-M5K | RE-M12K | **Average** |
|---|---|---|---|---|---|---|
| RetGK$_I$(Non) | 81.0±0.3 | 71.9±1.0 | 47.7±0.3 | **56.1±0.5** | **48.7±0.2** | **61.08** |
| RetGK$_{II}$(Non) | 80.6±0.3 | 72.3±0.6 | 48.7±0.6 | 55.3±0.3 | 47.1±0.3 | 60.8 |
| RetGK$_{II}$(MC)(Non) | 73.6±0.3 | 71.0±0.6 | 46.7±0.6 | 54.2±0.3 | 45.9±0.2 | 58.28 |
| GNTK | **83.6±1.0** | **76.9±3.6** | **52.8±4.6** | - | - | - |
| GCN | 81.72±1.64 | **73.30±5.29** | 51.20±5.13 | 56.81±2.37 | 49.31±1.44 | 62.47 |
| GLN | 75.72±2.51 | 73.10±3.18 | 50.40±5.61 | 52.97±2.58 | 39.84±0.95 | 58.41 |
| GFN | 81.50±2.42 | 73.00±4.35 | **51.80±5.16** | **57.59±2.40** | 49.43±1.36 | **62.66** |
| GFN-light | 81.34±1.73 | 73.00±4.29 | 51.20±5.71 | 57.11±1.46 | **49.75±1.19** | 62.48 |

## I  VARYING DATASET SIZE

To test the impact of dataset size, we take the largest graph dataset, RE-M12K, which has 11929 graphs. And we then construct nine new datasets by randomly sampling the original dataset with different ratios, ranging from 10% to 100% of all graphs. We compute both training and test accuracies over 10 fold cross-validation for both our GFN and GCN. For each dataset size (10 fold cross validation), we consider two ways to extract performance: (1) selecting best epoch averaged over 10 fold cross validation, or (2) selecting the last epoch (i.e. the 100-th epoch).

Figure 4 shows the results. We can see that as data size increases, 1) it is harder for both models to overfit (training accuracy decreases), but it seems GCN still overfits more if trained longer (to the last epoch); 2) at the best epoch, both models performance almost identical, without significant gaps between them.

## J  GRAPH VISUALIZATIONS

Figure 5, 6, 8, and 7 show the random and mis-classified samples for MUTAG, PROTEINS, IMDB-B, and IMDB-M, respectively. In general, it is difficult to find the patterns of each class by visually examining the graphs. And the mis-classified patterns are not visually distinguishable, except for IMDB-B/IMDB-M datasets where there are some graphs seem ambiguous.

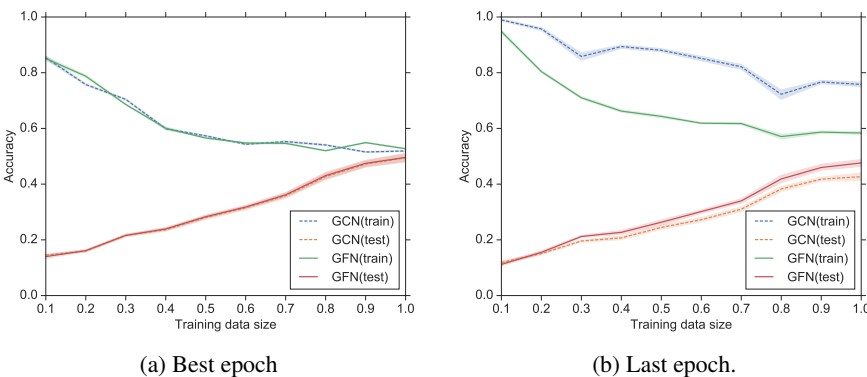

(a) Best epoch

(b) Last epoch.

Figure 4: Performances under varied dataset size. As dataset sizes increases, it becomes harder to overfit (especially for GFN), but GFN still performs as well as, if not better, than GCN.

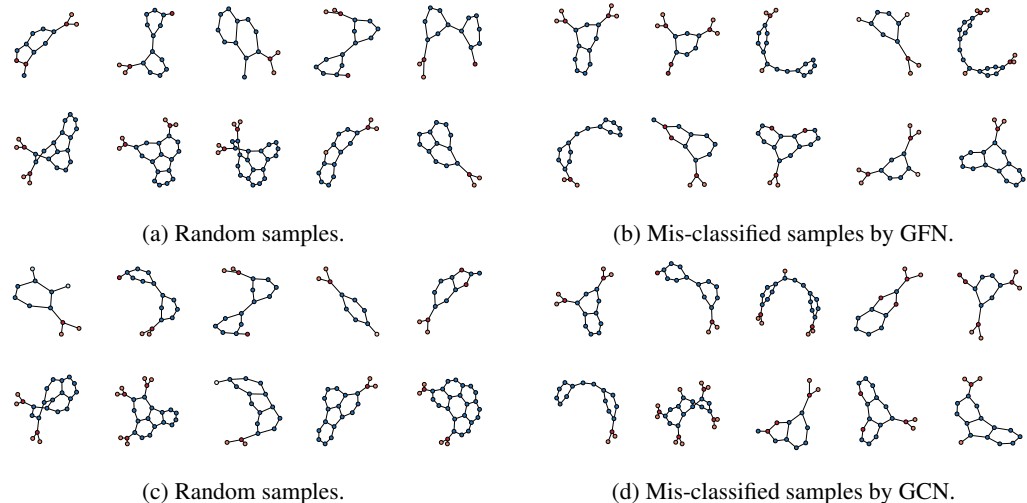

(a) Random samples.

(b) Mis-classified samples by GFN.

(c) Random samples.

(d) Mis-classified samples by GCN.

Figure 5: Random and mis-classified samples from MUTAG. Each row represents a (true) class.

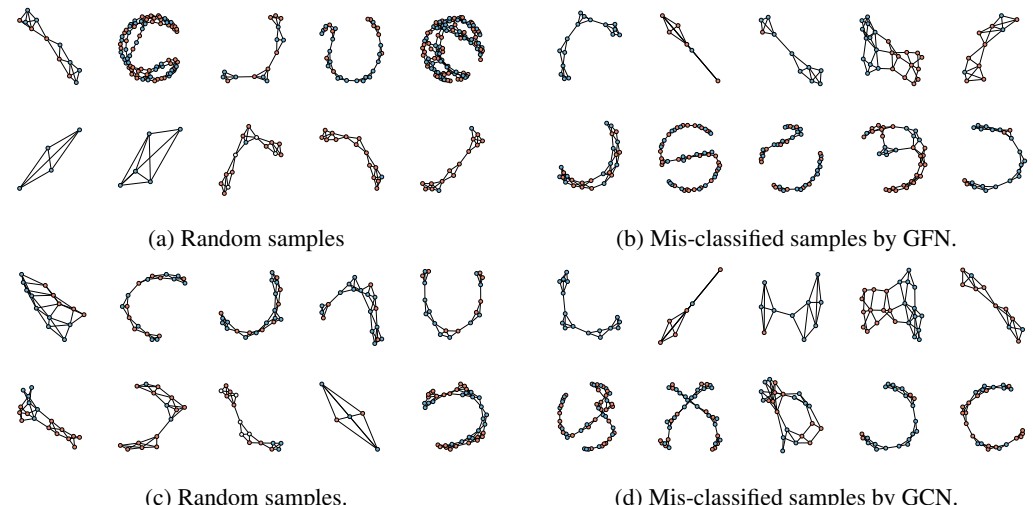

(a) Random samples

(b) Mis-classified samples by GFN.

(c) Random samples.

(d) Mis-classified samples by GCN.

Figure 6: Random and mis-classified samples from PROTEINS. Each row represents a (true) class.

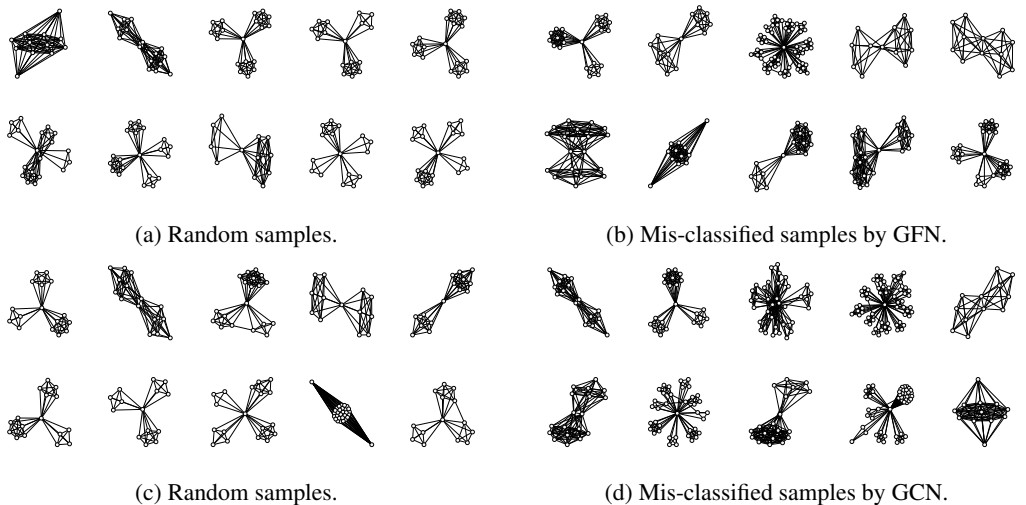

(a) Random samples.          (b) Mis-classified samples by GFN.

(c) Random samples.          (d) Mis-classified samples by GCN.

Figure 7: Random and mis-classified samples from IMDB-B. Each row represents a (true) class.

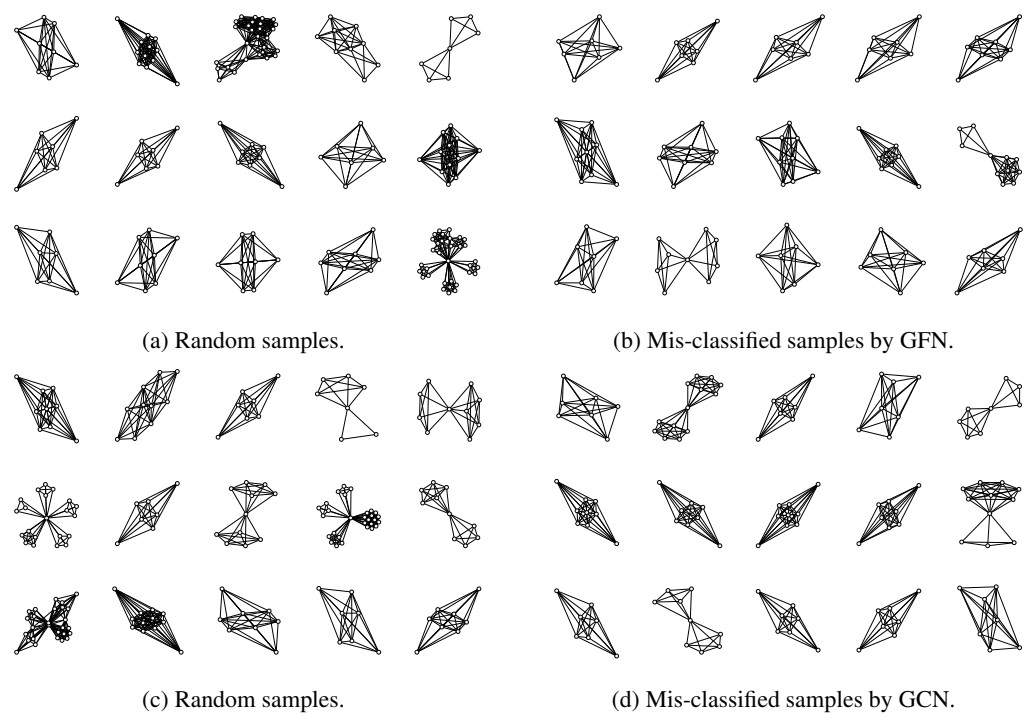

(a) Random samples.          (b) Mis-classified samples by GFN.

(c) Random samples.          (d) Mis-classified samples by GCN.

Figure 8: Random and mis-classified samples from IMDB-M. Each row represents a (true) class.

