# OpenReview forum: "Are Powerful Graph Neural Nets Necessary? A Dissection on Graph Classification"
_ICLR.cc/2020/Conference — Reject_

### Official Review · AnonReviewer1 · 2019-10-23
**Official Blind Review #1**

**Rating:** 3

**Review:**

This paper tries to study the importance of different components of GNNs. This paper studies two components 1) graph filtering: aggregation of neighboring features and 2) the aggregation function for the output.

To study this problem, this paper proposes two models, Graph Feature Network (GFN) and Graph Linear Network (GLN). GFN first uses the adjacency matrix to create several layers of features, then applies a multi-layer fully-connected neural network. GLN is a special case of GFN with the fully-connected neural network being linear.

This paper conducts experiments on graph classification task and finds GFN gives a reasonable performance, whereas GLN's performance is weaker.



Comments:
This paper studies an important problem in GNN, and the proposed method is interesting. However, I cannot accept the paper in the current form because of the following reasons.

1. There is no theoretical analysis in the paper. For example, on some datasets, GFN, GLN, and GNN's performances are close while on other datasets, there are gaps. The current paper does not provide insight.

2. GNN also contains non-linearity in the middle layers. However, the methodology in this paper cannot account for the importance of non-linearity in the middle layers.

3. The experiment section ignores some recent results on graph classification tasks. See:
https://arxiv.org/abs/1809.02670
https://arxiv.org/abs/1905.13192

**Experience Assessment:**

I have published one or two papers in this area.

**Review Assessment: Checking Correctness Of Derivations And Theory:**

I carefully checked the derivations and theory.

**Review Assessment: Checking Correctness Of Experiments:**

I carefully checked the experiments.

**Review Assessment: Thoroughness In Paper Reading:**

I read the paper thoroughly.

---

> ### Author Response · Authors · 2019-11-12
> **Response**
>
> We thank the reviewer for the time and detailed comments. Please find our responses to the comments below.
>
> [Analysis and insights]
>
> Two types of theoretical analysis are presented in this paper:
>     1) We prove that GFN can be derived by linearizing graph filtering part of GNNs (proposition 1), and leverage this theoretical connection to decouple the two GNN parts and study the importance of them separately.
>     2) We show that GFNs can be a very powerful framework without the restriction on the feature extraction function γ(G, X) and the exact forms of the set function (proposition 2), which is encouraging for future graph function design.
>
> Regarding the gaps between GCN and GFN among datasets, we note that 6 out of 10 datasets, GFN outperforms GNN counterpart in fair comparisons, and also note the gaps are *small* as they are within *1 standard deviation*. We are not convinced if these gaps are substantial, and thus conclude that both methods are on par across the whole set of benchmarks.
>
> The main insight of this paper is that linear graph filtering with non-linear set function is an efficient and powerful scheme for modeling existing graph classification benchmarks.
>
> [Non-linearity in GNN’s middle layers]
>
> We do account for the non-linearity in GNN’s as our GNN baselines have non-linearity in them. When nonlinearity in GNN’s middle layers are removed, we prove that they can be expressed as a GFN with appropriated graph features (in proposition 1). By comparing GFN and GNN, we are testing the importance of the nonlinearity of the graph filtering function (in GNN’s middle layers).
>
> [More comparisons]
>
> At the time when this work was conducted, the state-of-the-art of GNN variant was GIN (Xu et al ICLR’19), which we compared in this work (among other 7 baselines). We’d also like to point out that our goal is to dissect GNN variants, while both suggested papers are based on graph kernels, which are quadratic to the number of nodes and graphs (e.g. faster RetGKII is generally inferior than much slower RetGKI, and GNTK cannot scale to Reddit datasets). In contrast, our GFN has linear complexity thus in practice very fast/scalable (Figure 2), and the performances are better or comparable averaged over all benchmarks. Nonetheless, we appreciate the reviewer’s time and evaluation thus have added the full comparison and discussion in the revision (Appendix H), in good faith that the reviewer would also appreciate the contributions of our work.
>
> We’d like to clarify further as necessary, so please feel free to let us know if any of the concerns are not fully addressed.

---

### Official Review · AnonReviewer3 · 2019-10-24
**Official Blind Review #3**

**Rating:** 6

**Review:**

This paper presents a dissection analysis of graph neural networks by decomposing GNNs into two parts: a graph filtering function and a set function. Although this decomposition may not be unique in general, as pointed out in the paper, these two parts can help analyze the impact of each part in the GNN model. Two simplified versions of GNN is then proposed by linearizing the graph filtering function and the set function, denoted as GFN and GLN, respectively. Experimental results on benchmarks datasets for graph classification show that GFN can achieve comparable or even better performance compared to recently proposed GNNs with higher computational efficiency. This demonstrates that the current GNN models may be unnecessarily complicated and overkill on graph classification. These empirical results are pretty interesting to the research community, and can encourage other researchers to reflect on existing fancy GNN models whether it's worth having more complex and more computationally expensive models to achieve similar or even inferior performance. Overall, this paper is well-written and the contribution is clear. I would like to recommend a weak accept for this paper. If the suggestions below can be addressed in author response, I would be willing to increase the score.


Suggestions for improvement:

1) Considering the experimental results in this paper, it is possible that the existing graph classification tasks are not that difficult so that the simplified GNN variant can also achieve comparable or even better performance (easier to learn). This can be conjectured from the consistently better training performance but comparable testing performance of original GNN. Another possibility is that even the original GNN has larger model capacity, it is not able to capture more useful information from the graph structure, even on tasks that are more challenging than graph classification. However, this paper lacks such in-depth discussions;

2) Besides the graph classification task, it would be better to explore the performance of the simplified GNN on other graph learning tasks, such as node classification, and various downstream tasks using graph neural networks. This can help demystify the question raised in the previous point; 3) The matrix \tilde{A} in Equation 5 is not well explained (described as "similar to that in Kipf and Welling (2016)"). It would be more clear to directly point out that it is the adjacency matrix, as described later in the paper.

**Experience Assessment:**

I have published one or two papers in this area.

**Review Assessment: Checking Correctness Of Derivations And Theory:**

I assessed the sensibility of the derivations and theory.

**Review Assessment: Checking Correctness Of Experiments:**

I carefully checked the experiments.

**Review Assessment: Thoroughness In Paper Reading:**

I read the paper thoroughly.

---

> ### Author Response · Authors · 2019-11-12
> **Response**
>
> Thank you for your time and valuable feedbacks.
>
> [More discussion on the observations]
>
> We agree that there is more than one possibility for the empirical observations. Allow us to re-elaborate our main observation: what our experiments show is that GNN can overfit training set, but it doesn’t generalize better than GFN (GNN with linearized graph filtering function) on a broad set of benchmarks.
>
> One possibility is the inadequacy of existing graph classification benchmarks, which we are inclined to think it is the case. We have tried our best and tested on the most widely used benchmarks across the spectrum (from 188 to 11929 graphs). We also try to varying the dataset size by subsampling the largest dataset (RE-M12K), and the results can be found in appendix I. We hope, along with the whole community, to realize and adopt more complex real datasets to test if the observation still persists.
>
> The other possibility is that the linear graph filtering may be a good inductive bias for the tested datasets/problems. This is what other studies on node classification (e.g. Wu et al, ICML’19) suggest as well - GNNs are performing low-pass filtering. However, this is again dependant on the tasks and datasets considered.
>
> The third possibility is that, as suggested by the reviewer, despite GNNs can overfit but they are not capable of capturing the generalizable features (at least not prioritizing to learn those features). To show this is the case, we need to improve existing GNNs (e.g. architecture, objective) so that they can generalize better in existing benchmarks. We have not been able to find new techniques, or existing work, that can identify those more generalizable features.
>
> We admit that our work has limitations on fully answering these questions, but we believe raising the right question itself (with solid experimental observations) is an important step towards the good answers. We wish our work can raise the awareness of the phenomenon so that it can be better studied in the future. What’s more, the proposed GFN can serve as a fast and accurate approximation to GNN for graph classification task, which we believe is a practical contribution.
>
> [Other datasets and tasks]
>
> In this work we focus on graph classification problem on 12 datasets as they are the most widely used benchmarks for recently proposed advanced GNN variants. However, we agree that to  further demystify the above possibilities, more work should be done to adopt GFN as baseline and apply it for more datasets/tasks.
>
> Toward that end, we conduct experiments on image classification as graph classification on MNIST, where we find significant gap between GCN and GFN (Appendix D), which suggests non-linear graph filtering is important for image classification by treating images as graphs (unlike other natural graph datasets). We wish to conduct more meaningful downstream tasks that use graph neural nets but it requires careful selection and establishing benchmark datasets, thus we defer it for future work.
>
> As for node classification task, which does not require the graph readout function (i.e., only has graph filtering function), and typically it is tested in the transductive setting (i.e. single graph), thus is a simpler task. Wu et al (ICML’19) has shown that fully linearizing GCN yields similar performance in several node classification datasets, which can be seen as a special case of GFN without set function. However, fully linearizing GCN for graph classification (i.e. GLN) significantly degenerates the performance, making it an important distinction between graph and node classification tasks.
>
> [Notation clarification]
>
> We have updated the draft to clarify \tilde{A}. Shown below Eq 2, $\tilde{A} = \tilde{D}^{-1/2}(A+\epsilon I)\tilde{D}^{-1/2}$ is the normalized adjacency matrix, with \epsilon=1 this is the one proposed in Kipf and Welling (2016). We use this formulation by default (with \epsilon=1e-8), but want to note that other formulation of \tilde{A} is also allowed (e.g. different normalized graph Lapacian) under the general framework of GFN.

---

### Official Review · AnonReviewer2 · 2019-10-26
**Official Blind Review #2**

**Rating:** 6

**Review:**

The paper dissects the importance of two parts in GCN: 1) nonlinear neighborhood aggregation; 2) nonlinear set function by linearizing the two parts and resulting in Graph Feature Network (GFN) and Graph Linear Network (GLN). It shows empirically that GFN achieves almost the same performance while GLN is much worse, suggesting the nonlinear graph neighborhood aggregation step may be unnecessary. Extensive ablation studies are conducted to single out the effects of various factors.

The paper studies an interesting problem and sets out a good plan of experiments to verify the hypotheses. The results are interesting: merely constructing graph neighborhood features alone is enough to get comparable performance with GCN since the nonlinearity in the set function is strong enough. The experiments are designed nicely: 1) it compares with various baselines on a variety of popular benchmarks; 2) ablation studies single out the importance of different graph features, such as degree, and multi-hop averages; 3) verifying whether the good performance GFN comes from easier optimization.

The paper is also clearly written, with clean notations, and well-structured sections.

I think the experiment can be improved by comparing on larger, more complex datasets. Figure 1 seems to suggest GCN is overfitting compared to GFN due to its extra capacity--significantly better training accuracy but slightly worse test accuracy. It is usually the case that larger and more complex datasets require more sophisticated models. But the paper makes a good case for GFN in these datasets for the graph classification task.

**Experience Assessment:**

I have read many papers in this area.

**Review Assessment: Checking Correctness Of Derivations And Theory:**

I carefully checked the derivations and theory.

**Review Assessment: Checking Correctness Of Experiments:**

I carefully checked the experiments.

**Review Assessment: Thoroughness In Paper Reading:**

I read the paper at least twice and used my best judgement in assessing the paper.

---

> ### Author Response · Authors · 2019-11-12
> **Response**
>
> Thank you for your time and positive feedbacks. Regarding the datasets, we compared 12 social and biological graph datasets (from 188 to 11929 graphs), which are the most widely used standard benchmarks for graph classification task as of today (some existing work does not even include the largest RE-M12K due to scalability issue).
>
> On the dataset size, we try to incorporate your comments and perform extra experiments. To see how the varying dataset size affects the performance of GFN and GCN, we take the largest RE-M12K dataset (11929 graphs), and randomly sample datasets of different size (from 10% graphs to 100% graphs). We run 10 fold cross validation on each of the dataset, and found that: as dataset sizes increases, it becomes harder to overfit (especially for GFN), but GFN still performs as well as, if not better, than GCN. Details of this experiment are added to the appendix I.
>
> On the dataset complexity, we fully agree that with more complex tasks/datasets powerful GNNs could probably show better performance. And in fact, that is also part of our goal in publishing our work, to raise the awareness that common graph classification benchmarks are likely inadequate for testing advanced GNN variants. We wish the community as a whole to explore and adopt more convincing benchmarks for testing advanced GNN variants, or include GFN as a standard baseline to provide a sanity check.

---

### Public Comment · ~Boris_Knyazev1 · 2019-10-31
**Relation to Chebyshev graph convolution**

Thank you for an interesting paper. I found that Eq. 5 resembles the Chebyshev graph convolution proposed in [1] which you cite in the first sentence only. The nice thing about [1] is that it approximates spectral graph convolution if K is large enough and uses the orthogonal Chebyshev basis, so it's theoretically sound. In your Eq. 5 you just take powers of adjacency matrices, and my hypothesis is that it can lead to unstable training dynamics, which might explain somewhat lower than expected performance when you use K>1.

It would be interesting to see the connection of your formulation in Eq. 5 to [1].

[1] Convolutional Neural Networks on Graphs with Fast Localized Spectral Filtering

---

> ### Author Response · Authors · 2019-11-01
> **It is one instantiation of graph augmented features in our framework**
>
> Thanks for your interest in our work. It is fair to instantiate graph augmented features with other filters/operators, in our work, we follow Kipf and Welling (2016) and use modified adjacency matrix with renormalization trick, which is shown to be better than Chebyshev polynomials in their work. But I think Chebyshev polynomials can probably be used as another instantiation of the graph augmented features, along with possibly many more. We are different from those GCN based methods (with normalized adj or Chebyshev polynomials) in the sense that we fix graph augmented features in learning time, and treat the graph as a set.

---

### Author Response · Authors · 2019-11-12
**Revision log**

This is to log what we have changed in the revision:

1. We added comparisons to RETGK and GNTK as suggested by Reviewer 1.
2. We clarified a notation as suggested by Reviewer 2.
3. We added an experiment on varying dataset size according to the comment of Reviewer 3.

---

### Public Comment · ~Clément_Vignac1 · 2020-03-12
**Related results for graph classification**

Hello,
I just discovered your paper, thank you for this very comprehensive work. I wanted to bring to your attention a short paper about node classification, in which we reached very similar conclusions: https://arxiv.org/pdf/1911.05384.pdf

Our main focus was the study of the relative performance of graph neural networks depending on the number of training examples and features in the dataset, which is a different perspective. However, one observation that we made is that even when a lot of training data is available, intertwining propagation and learning layers is not useful. We found that in this case, it was better to use several propagation layers (with no trainable parameters), followed by a non-linear feature extractor (i.e a MLP). It looks very close to your conclusions about graph classification.

We were not aware of your work when we wrote the paper and therefore did not cite you, but we'll make sure to do it whenever we present it.

---

### Decision · Program_Chairs · 2019-12-19

**Decision:**

Reject

**Comment:**

This paper proposes to split the GNN operations into two parts and study the effects of each part. While two reviewers are positive about this paper, the other reviewer R1 has raised some concerns. During discussion, R1 responded and indicated that his/her concerns were not addressed in author rebuttal. Overall, I feel the paper is borderline and lean towards reject.